# TimeExpert: Boosting Long Time Series Forecasting with Temporal Mix of Experts

## Abstract

Transformer-based architectures dominate time series modeling by enabling global attention over all timestamps, yet their rigid "one-size-fits-all" context aggregation fails to address two critical challenges in real-world data: (1) inherent lag effects, where the relevance of historical timestamps to a query varies dynamically; (2) anomalous segments, which introduce noisy signals that degrade forecasting accuracy. To resolve these problems, we propose the Temporal Mix of Experts (TMOE)—a novel attention-level mechanism that reimagines key-value (K-V) pairs as local experts (each specialized in a distinct temporal context) and performs adaptive expert selection for each query via localized filtering of irrelevant timestamps. Complementing this local adaptation, a shared global expert preserves the Transformer's strength in capturing long-range dependencies. We then replace the vanilla attention mechanism in popular time-series Transformer frameworks (i.e., PatchTST and Timer) with TMOE, without extra structural modifications, yielding our specific version TimeExpert and general version TimeExpert-G. Extensive experiments on seven real-world long-term forecasting benchmarks demonstrate that TimeExpert and TimeExpert-G outperform state-of-the-art methods. Code will be released after acceptance.

## 1 Introduction

Time series forecasting is a fundamental task across diverse real-world domains, encompassing scenarios such as energy grid management, urban transportation planning, climate science, and environmental monitoring (Bi et al., 2023; Truong et al., 2022; Kadiyala & Kumar, 2014; Kardakos et al., 2013). The core challenge lies in capturing complex temporal dependencies, such as short-term cyclical fluctuations, long-range trend correlations, and sudden disturbances from anomalous events, while simultaneously ensuring the efficient processing of long input sequences. In recent years, Transformer-based architectures (Zhou et al., 2021; 2022; Wu et al., 2021) have revolutionized time series modeling through their self-attention mechanism. Unlike Recurrent Neural Networks (RNNs) (Lai et al., 2018; Rasul et al., 2021), which process timestamps sequentially (limiting long-range dependency capture), or Convolutional Neural Networks (CNNs) (Liu et al., 2022; Wu et al., 2022), which rely on fixed local receptive fields (limiting adaptive context aggregation), Transformers simultaneously attend to all timestamps in a sequence. This global attention mechanism has enabled Transformers to achieve state-of-the-art (SOTA) performance on benchmarks spanning power grid, transportation, and meteorological data, establishing them as the de facto standard for time series forecasting.

However, despite their success, Transformer-based models face inherent limitations when applied to real-world time-series data—challenges that the "one-size-fits-all" nature of their attention mechanism fails to address, as shown in Figure 1. First, inherent lag effects characterize most time series: the relevance of historical timestamps to a query timestamp varies dynamically. For example, in traffic forecasting, a morning rush-hour query timestamp may depend strongly on timestamps from the previous 30 minutes (short-range lag) on weekdays but on timestamps from the prior weekend (long-range lag) during holidays. Forcing all timestamps to attend to the same global context ignores this time-varying relevance, leading to suboptimal weighting of critical historical information. Second, anomalous segments such as sensor glitches in electricity data or accidents in traffic data may introduce noisy, misleading signals into the sequence. Global attention mechanisms, which aggregate information from all timestamps, inadvertently incorporate these anomalies into the context,

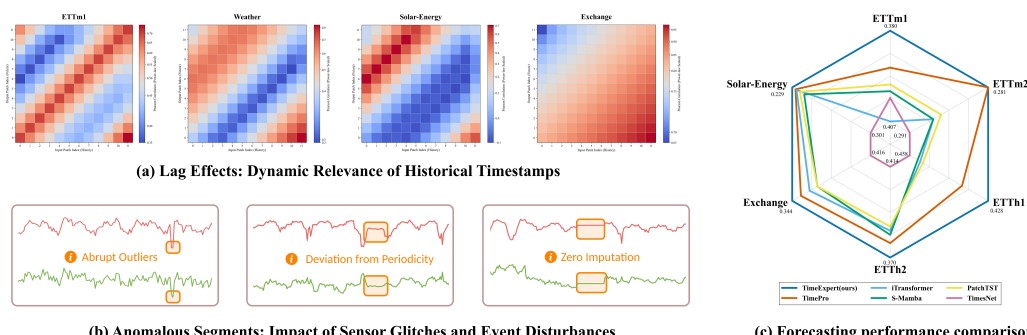

(a) Lag Effects: Dynamic Relevance of Historical Timestamps

(b) Anomalous Segments: Impact of Sensor Glitches and Event Disturbances

(c) Forecasting performance comparison

Figure 1: Illustration of temporal forecasting challenges and performance. (a) Heatmaps of average Pearson correlation coefficients between input and output patches, showing lag effects in historical timestamps. (b) Examples of anomalous time series segments, including abrupt outliers, periodicity deviations, and zero imputation artifacts. (c) Radar chart comparing forecasting performance (MSE) of our method and state-of-the-art baselines across multiple benchmarks.

degrading forecasting accuracy. These two challenges collectively highlight a key gap: the rigid global context aggregation of existing Transformers is misaligned with the dynamic, noisy nature of real-world time series.

Prior work has attempted to address these limitations, but with notable shortcomings. On the one hand, localized attention variants (Lin et al., 2021; Zhou et al., 2021) introduce convolutional layers or fixed sliding windows to focus on local context, mitigating anomaly interference but sacrificing the Transformer's strength in long-range dependency capture. On the other hand, Mixture-of-Experts (MOE) models for time series (Shi et al., 2025) use sparse expert layers to replace feed-forward networks (FFNs), enabling specialization for multi-frequency data but treating experts as layer-level components rather than context-level adapters, which fails to address the root issue of irrelevant timestamp filtering for each query. Neither line of work resolves the dual need for adaptive local context selection (to handle lags and anomalies) and preserved global awareness.

Motivated by this gap, we propose Temporal Mix of Experts (TMOE), a novel attention-level mechanism that reimagines the core building blocks of Transformer self-attention to balance local adaptation and global awareness. TMOE's key innovation lies in treating key-value (K-V) pairs as local experts, each corresponding to a distinct historical timestamp and specializing in encoding its associated temporal context. For every query timestamp, TMOE performs localized filtering of irrelevant context timestamps: a learnable gating function scores each local expert based on its relevance to the query (e.g., via cosine similarity and temporal proximity) and selects only the top-k most relevant experts for context aggregation. This adaptive selection directly mitigates the interference of anomalous segments (by down-weighting experts associated with noisy timestamps) and aligns with dynamic lag effects (by prioritizing experts from time-varying relevant intervals). To complement this local adaptation and avoid losing the Transformer's global advantage, TMOE further introduces a shared global expert that is integrated into the context aggregation of every query timestamp and trained to encode long-range sequence trends.

We seamlessly integrate the TMOE mechanism into the standard Transformer framework without modifying its overall structure (e.g., retaining the encoder-decoder pipeline for forecasting in PatchTST (Nie et al., 2023) and Timer (Liu et al., 2024b)), resulting in our proposed specific- and general-version model: TimeExpert and TimeExpert-G. This integration ensures compatibility with existing Transformer-based forecasting pipelines while enhancing adaptability to real-world data. To validate TimeExpert's effectiveness, we conduct extensive experiments on seven widely used real-world long-term forecasting benchmarks, covering energy (Solar), exchange, environmental (Weather) domains, and a general electricity dataset (ETT). Our results demonstrate that TimeExpert and TimeExpert-G outperform SOTA baselines including PatchTST (Nie et al., 2023), Timer (Liu et al., 2024b), and Time-MOE (Shi et al., 2025) by 3.2–7.8% in mean absolute error (MAE) and 2.9–6.5% in mean squared error (MSE). Notably, the TimeExpert family maintains compara-

ble computational overhead to baseline Transformers, as the localized expert selection reduces the number of attention computations despite the addition of the global expert.

The main contributions of this work are summarized as follows:

- A novel TMOE mechanism: We redefine K-V pairs as local experts and introduce adaptive expert selection via localized context filtering, addressing the limitations of rigid global attention in handling lag effects and anomalies. In addition, a shared global expert is introduced to preserve long-range dependency capture, balancing the strengths of localized adaptation and global context awareness.

- A powerful and elegant TimeExpert family: We design the TimeExpert family including specific version TimeExpert and general version TimeExpert-G by directly integrating TMOE into the popular time-series Transformer frameworks. It is compatible with the Transformer architecture, simple in structure, and easy to scale.

- State-of-the-Art Performance: TimeExpert and TimeExpert-G achieve SOTA results on seven real-world long-term forecasting benchmarks, with empirical evidence validating the effectiveness of TMOE via qualitative and quantitative analysis.

## 2 RELATED WORK

### 2.1 LONG-TERM TIME-SERIES FORECASTING MODELS

Capturing long-range dependencies remains a key challenge in Long-term Time-Series Forecasting (LTSF). Early deep learning models such as Recurrent Neural Networks (RNNs) (Hochreiter & Schmidhuber, 1997; Chung et al., 2014; Greff et al., 2016; Sherstinsky, 2020) struggled with this due to their sequential nature. The introduction of the Transformer (Vaswani et al., 2017) brought a breakthrough, enabling global dependency modeling. However, its standard self-attention mechanism, designed for general-purpose tasks, does not fully exploit the characteristics of time-series data. To address this, recent research has followed three main directions. The first aims to reduce the cost of full attention through sparse or low-rank approximations (e.g., Linformer (Wang et al., 2020), Longformer (Beltagy et al., 2020)). The second introduces inductive biases specific to time series, such as the Autoformer (Wu et al., 2021)'s autocorrelation mechanism to model seasonality. The third, now widely adopted, is the patching paradigm. Models like PatchTST (Nie et al., 2023) and iTransformer (Liu et al., 2024a) divide long sequences into non-overlapping patches and apply attention across patch tokens, achieving strong performance.

Despite their effectiveness, these patching-based models typically apply uniform attention over all patches, assuming equal importance across the sequence. We argue this is a key limitation: irrelevant patches may introduce noise and degrade representation quality. Rather than focusing on how attention is computed, our work shifts the focus to what attention should compute over. We propose a selective attention mechanism that first identifies the most relevant patch interactions, and only then applies attention to this filtered subset. By emphasizing truly informative signals, this approach improves modeling precision and ultimately achieves better forecasting accuracy.

### 2.2 MIXTURE-OF-EXPERTS (MOE)

Mixture-of-Experts (MoE) (Jacobs et al., 1991; Jordan & Jacobs, 1994) is a conditional computation framework that scales model capacity at a near-constant computational cost. Its modern revival has been driven by the Sparsely-Gated MoE with Top-K routing, a key technology for scaling large models (Lepikhin et al., 2020; Fedus et al., 2022; Du et al., 2022) in NLP and vision. Despite its immense success in NLP and vision, the application of MoE in time-series forecasting is still nascent. The few existing works, such as Time-MoE (Shi et al., 2025), which assign different experts to trend and seasonal components, typically employ MoE in a static, sequence-level manner, lacking the fine-grained, dynamic routing capabilities of modern MoE architectures. In this paper, we propose a dynamic and fine-grained Top-k MoE framework for LTSF that treats each time series patch as an individual token for dynamic routing by a routing network. This patch-level dynamic routing enables the model to invoke specialized experts to handle diverse temporal patterns, including trends and anomalies.

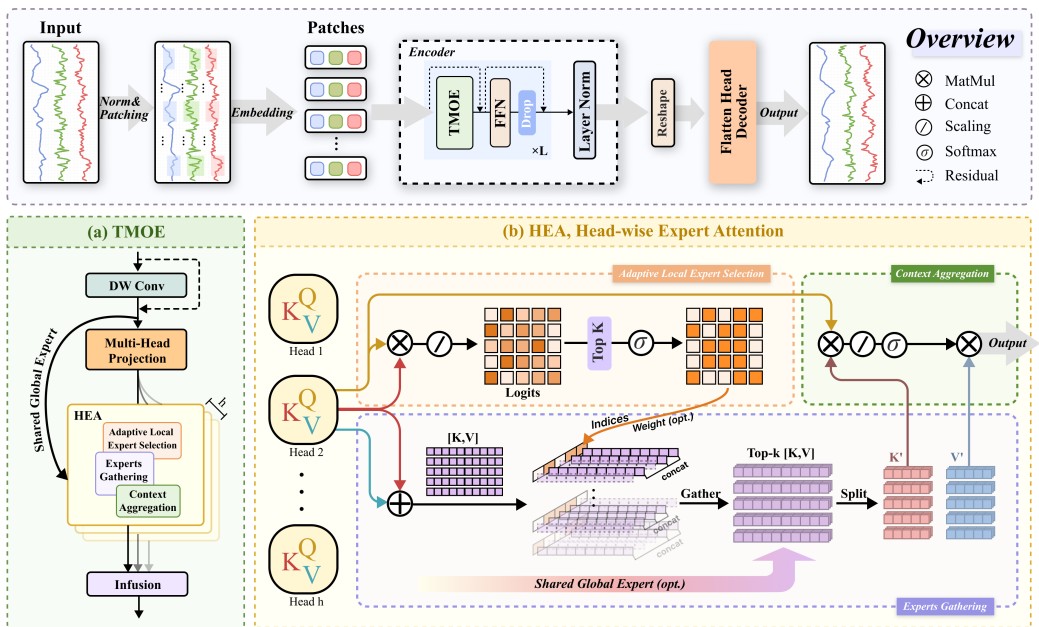

Figure 2: Overview of our TimeExpert method, which primarily consists of three components: patch embedding, encoder, and flatten head decoder. We employ TMOE in the encoder instead of vanilla attention to model the time-varying correlations between timestamps caused by lag effects and reduce interference from anomalous segments.

# 3 METHODOLOGY

## 3.1 PRELIMINARIES

We first review vanilla Transformer self-attention to establish notation and context. We employ a channel-independent approach to perform identical attention operations on each channel. Therefore, for simplicity, all subsequent operations are implemented exclusively on single-channel sequences. For time series forecasting, let the input sequence be $X = [x_1, x_2, ..., x_T] \in \mathbb{R}^{T \times D}$, where $T$ is the sequence length and $D$ is the feature dimension of each timestamp $x_t$. The Transformer's self-attention layer maps $X$ to a context-rich output $Y = [y_1, y_2, ..., y_T] \in \mathbb{R}^{T \times D}$ by computing $Q = XW_Q, K = XW_K$, and $V = XW_V$ projections, followed by scaled dot-product attention,

$$\alpha_{t,s} = \frac{\exp\left(\frac{q_t \cdot k_s^\top}{\sqrt{d_k}}\right)}{\sum_{s'=1}^{T} \exp\left(\frac{q_t \cdot k_{s'}^\top}{\sqrt{d_k}}\right)}, \tag{1}$$

where $Q, K, V \in \mathbb{R}^{T \times d_k}$, with $q_t \in \mathbb{R}^{d_k}$, $k_t \in \mathbb{R}^{d_k}$, $v_t \in \mathbb{R}^{d_k}$ denoting the query, key, and value vectors for timestamp $t$. $\alpha_{t,s} \in [0, 1]$ represents the relevance of timestamp $s$ to query $t$, and the softmax ensures weights sum to 1. The context vector $y_t$ for query $t$ is then the weighted sum of all values: $y_t = \sum_{s=1}^{T} \alpha_{t,s} v_s$. As highlighted in the Introduction, this "one-size-fits-all" standard formulation suffers from two critical flaws for real-world time series:

1) Inherent lag effects: All queries $q_t$ share the same content-independent key-value pairs $\{K, V\}$, ignoring the time-varying correlations caused by lag effects.

2) Anomalous segments: The dot-product $q_t \cdot k_s^\top$ captures feature similarity among all $T$ timestamps, and thereby inadvertently incorporates these anomalies into the context, degrading forecasting accuracy.

Therefore, we propose the Temporal Mix of Experts (TMOE) to address these limitations by redefining how K-V pairs contribute to context aggregation, as detailed next.

## 3.2 Temporal Mix of Experts

The core innovation of TMOE is to treat key-value (K-V) pairs as local experts—each specialized in encoding the temporal context of its associated timestamp—and to enable adaptive expert selection for each query. A shared global expert further preserves long-range dependency capture.

**Local Expert Definition** For each historical timestamp $s$, we define a local expert $e_s = (k_s, v_s)$, where $k_s$ (key) encodes the feature context of $s$, and $v_s$ (value) encodes the predictive information of $s$. Unlike prior MOE work (e.g., MOIRAI-MOE) that defines experts as feed-forward layers, TMOE's local experts are context-level adapters, that is, each $e_s$ is tied to a specific timestamp, enabling fine-grained filtering of irrelevant temporal information.

**Adaptive Local Expert Selection** For each query $q_t$, we select a subset of local experts that are most relevant to $t$, filtering out experts associated with anomalies or mismatched lags. To capture both feature similarity (like standard attention) and temporal relevance (to model lag effects), we compute a combined score $s_{t,s}$ for expert $e_s$ and query $q_t$:

$$s_{t,s} = \underbrace{\frac{q_t \cdot k_s^\top}{\sqrt{d_k}}}_{\text{Feature Similarity}} \cdot \underbrace{\psi\left(|t-s|\right)}_{\text{Temporal Relevance}} \tag{2}$$

Here, $\psi$ is a learnable projection function that controls the decay of relevance with temporal distance $|t - s|$ for long- or short-lag scenarios. In addition, this score naturally downweights experts from anomalous timestamps: if $x_s$ is an outlier, $k_s$ will be dissimilar to $q_t$, reducing $s_{t,s}$. To avoid over-parameterization and ensure efficiency, we select the top-$k$ experts with the highest scores $s_{t,s}$ (denoted $\mathcal{S}_t = \{e_s \mid s_{t,s} \in \text{top-}k(s_{t,1}, ..., s_{t,T})\}$). The size $k$ is a hyperparameter.

**Shared Global Expert** While local expert selection enhances adaptability, it risks losing the Transformer's strength in capturing long-range dependencies (e.g., seasonal trends spanning months). To mitigate this, we introduce a shared global expert $e_{\text{global}} = (k_{\text{global}}, v_{\text{global}})$, where:

$$k_{\text{global}} = v_{\text{global}} = \phi_{\text{global}}\left(\frac{\exp(x_{t,d})}{\sum_{t \in \top} \exp(x_{t,d})}\right), \quad \forall d \in d_k, \quad and \quad s_{t,global} = \frac{q_t k_{global}^T}{\sqrt{d_k}}, \tag{3}$$

where $\phi_{\text{global}}$ is a learnable projection, $k_{\text{global}} \in \mathbb{R}^{d_k}$ is the learnable vector encoding global sequence context (e.g., seasonal patterns). Then we update the selected set of experts to $\hat{S}_t = \{e_s \mid s_{t,s} \in \text{top-}k(s_{t,1}, ..., s_{t,T}), k_{\text{global}}\}$. The global expert is included in the context aggregation for every query, ensuring that even after local filtering, TMOE retains awareness of long-range trends.

**TMOE Context Aggregation** For the selected experts in $\hat{S}_t$, we normalize their scores via softmax to get gating weights:

$$g_{t,s} = \frac{\exp(s_{t,s})}{\sum_{e_{s'} \in \mathcal{S}_t} \exp(s_{t,s'})}, \quad \forall e_s \in \hat{\mathcal{S}}_t, \quad and \quad y_t^{\text{TMOE}} = \sum_{e_s \in \mathcal{S}_t} g_{t,s} v_s, \tag{4}$$

where $g_{t,s} \in [0, 1]$ and $\sum_{e_s \in \mathcal{S}_t} g_{t,s} = 1$. Note that we adopt the multi-head design, which enhances the diversity of selected experts and thereby improves forecasting performance.

## 3.3 TimeExpert and TimeExpert-G workflow

TimeExpert integrates the TMOE mechanism into the standard Transformer encoder framework (Nie et al., 2023) for long-term time series forecasting. As shown in Figure2, the end-to-end workflow proceeds in three stages: input processing, encoder with TMOE and decoder with flatten head. Specifically, the univariate time series is first embedded along the temporal dimension with the normalization and patching operation to get the series representation. Note that for multivariate time series, we process each variable equally along the channel-independent approach. Then the

Table 1: Multivariate long-term forecasting results with horizon $H \in \{96, 192, 336, 720\}$ and fixed lookback window length L = 96. Results are averaged from all prediction horizons.

| Models | TimeExpert (ours) | | TimePro (2025) | | S-Mamba (2025) | | SOFTS (2024) | | iTransformer (2024b) | | PatchTST (2023) | | Crossformer (2023) | | TiDE (2023) | | TimesNet (2023) | | DLinear (2023) | | SCINet (2022) | |
|---|---|---|---|---|---|---|---|---|---|---|---|---|---|---|---|---|---|---|---|---|---|---|
| Metric | MSE | MAE | MSE | MAE | MSE | MAE | MSE | MAE | MSE | MAE | MSE | MAE | MSE | MAE | MSE | MAE | MSE | MAE | MSE | MAE | MSE | MAE |
| ETTm1 | 0.380 | 0.396 | 0.391 | 0.400 | 0.398 | 0.405 | 0.393 | 0.403 | 0.407 | 0.410 | 0.396 | 0.406 | 0.513 | 0.496 | 0.419 | 0.419 | 0.400 | 0.406 | 0.403 | 0.407 | 0.485 | 0.481 |
| ETTm2 | 0.281 | 0.326 | 0.281 | 0.326 | 0.288 | 0.332 | 0.287 | 0.330 | 0.288 | 0.332 | 0.287 | 0.330 | 0.757 | 0.610 | 0.358 | 0.404 | 0.291 | 0.333 | 0.350 | 0.401 | 0.571 | 0.537 |
| ETTh1 | 0.428 | 0.432 | 0.438 | 0.438 | 0.455 | 0.450 | 0.449 | 0.442 | 0.454 | 0.447 | 0.453 | 0.446 | 0.529 | 0.522 | 0.541 | 0.507 | 0.458 | 0.450 | 0.456 | 0.452 | 0.747 | 0.647 |
| ETTh2 | 0.370 | 0.399 | 0.377 | 0.403 | 0.381 | 0.405 | 0.373 | 0.400 | 0.383 | 0.407 | 0.385 | 0.410 | 0.942 | 0.684 | 0.611 | 0.550 | 0.414 | 0.427 | 0.559 | 0.515 | 0.954 | 0.723 |
| Exchange | 0.344 | 0.394 | 0.352 | 0.399 | 0.367 | 0.408 | 0.361 | 0.402 | 0.360 | 0.403 | 0.367 | 0.404 | 0.940 | 0.707 | 0.370 | 0.413 | 0.416 | 0.443 | 0.354 | 0.414 | 0.750 | 0.626 |
| Weather | 0.251 | 0.276 | 0.251 | 0.276 | 0.251 | 0.276 | 0.255 | 0.278 | 0.258 | 0.278 | 0.256 | 0.279 | 0.259 | 0.315 | 0.271 | 0.320 | 0.259 | 0.287 | 0.265 | 0.317 | 0.292 | 0.363 |
| Solar-Energy | 0.229 | 0.263 | 0.232 | 0.266 | 0.240 | 0.273 | 0.229 | 0.256 | 0.233 | 0.262 | 0.236 | 0.266 | 0.641 | 0.639 | 0.347 | 0.417 | 0.301 | 0.319 | 0.330 | 0.401 | 0.282 | 0.375 |

time-varying correlations between timestamps are captured by multiple layers of Encoder blocks. The core component of Encoder block is TMOE, which employs adaptive local experts and shared global experts to mitigate the impact of irrelevant or anomalous timestamps on modeling correlations. Finally, the context-aggregated temporal representation is input to the flatten head, which outputs the prediction sequence through a simple projection mapping. Similarly, we replace the vanilla attention module in Timer (Liu et al., 2024b) with TMOE, thereby proposing TimeExpert-G. Following Timer, we also incorporate causal masking into TMOE. All other components remain consistent.

## 4 EXPERIMENTS

We conduct extensive experiments on 7 widely used real-world datasets from diverse application domains, including ETT (ETTh1, ETTh2, ETTm1, ETTm2), Exchange, Weather, and Solar-Energy. Detailed settings and additional experiments can be found in the appendix.

### 4.1 MAIN RESULTS

First, we report the results of the long-term forecasting in Table 1. As presented, TimeExpert achieves 13 out of 14 best results across 7 benchmark datasets, demonstrating comprehensive state-of-the-art (SOTA) performance. To further substantiate these results, we categorize our results and analysis into three representative scenarios: datasets with strong periodicity (ETT), datasets with low signal-to-noise ratio (Exchange), and datasets with single dominant periodicity (Solar-Energy).

**Complex Periodicity and Trends (ETT Datasets)** On ETTm1, TimeExpert reduces MSE by 2.8% compared to TimePro and 6.6% compared to iTransformer; on ETTh1, the reductions are 2.3% and 5.7%, respectively. This advantage is driven by the TMOE, which facilitates explicit temporal alignment. By sparsely selecting the most similar historical patches along the time axis, it directly models relationships between key cyclical events and bypasses the information dissipation risk of implicit state models (TimePro) and the indirectness of variable-level attention (iTransformer).

**Low Signal-to-Noise Ratio (Exchange Dataset)** On the Exchange characterized by high randomness, TimeExpert achieves an MSE of 0.344, a notable improvement over iTransformer (0.360) and TimePro (0.352). It stems from the inherent noise-filtering property of the Adaptive Local Expert Selection within TMOE. By focusing on significant underlying patterns, TimeExpert effectively disregards transient noise. This contrasts with methods that process the entire sequence, which risk integrating stochastic fluctuations into their representations and degrading forecast accuracy.

**Single-Dominant Periodicity (Solar-Energy Dataset)** On Solar-Energy, which exhibits a highly stable daily cycle, TimeExpert also shows competitive performance, surpassing TimePro but slightly behind the frequency-domain expert model SOFTS. This result indicates the trade-off between generalization and specialization: while SOFTS benefits from strong inductive bias toward ideal periodic signals, TimeExpert closely approaches its performance without any task-specific modifications. This underscores the universality of its core mechanism and its strength as a general-purpose foundation model for time series forecasting.

Furthermore, TimeExpert-G, as its generalized counterpart, highlights the framework's robustness and generalization capability. While TimeExpert is tuned for peak performance in this specific task,

Table 2: Full results of zero-shot forecasting experiments with horizon $H \in \{96, 192, 336, 720\}$ and fixed lookback window length L = 96. Results are averaged from all prediction horizons. A dash ('-') denotes datasets included in the model's pretraining and therefore excluded from testing.

| Models | TimeExpert-G | | Timer (2024) | | MOIRAI (2024) | | Chronos (2024) | | TimesFM (2024) | | Time-MoE (2025) | |
|---|---|---|---|---|---|---|---|---|---|---|---|---|
| Metric | MSE | MAE | MSE | MAE | MSE | MAE | MSE | MAE | MSE | MAE | MSE | MAE |
| ETTm1 | **0.345** | **0.387** | 0.768 | 0.568 | 0.390 | 0.389 | 0.551 | 0.453 | 0.435 | 0.418 | 0.376 | 0.406 |
| ETTm2 | **0.265** | 0.336 | 0.315 | 0.356 | 0.276 | **0.320** | 0.293 | 0.331 | 0.347 | 0.360 | 0.315 | 0.365 |
| ETTh1 | **0.386** | **0.417** | 0.562 | 0.483 | 0.510 | 0.469 | 0.533 | 0.452 | 0.479 | 0.442 | 0.394 | 0.420 |
| ETTh2 | 0.346 | 0.392 | 0.370 | 0.400 | 0.354 | **0.377** | 0.392 | 0.397 | 0.400 | 0.403 | 0.403 | 0.415 |
| Exchange | **0.342** | **0.394** | 0.392 | 0.425 | 0.385 | 0.417 | 0.370 | 0.412 | 0.390 | 0.417 | 0.432 | 0.454 |
| Weather | **0.212** | **0.256** | 0.292 | 0.313 | 0.260 | 0.275 | 0.288 | 0.309 | - | - | 0.270 | 0.300 |
| Solar-Energy | **0.358** | 0.372 | 0.771 | 0.604 | 0.714 | 0.704 | 0.393 | **0.319** | 0.500 | 0.397 | 0.411 | 0.428 |

TimeExpert-G's results are equally compelling. As observed in Table 2, it consistently secures the best or second-best performance across nearly all evaluations. This consistency at a top-tier level across diverse datasets is a strong testament to its value as a foundation model. This exceptional generalization, reflected in its state-of-the-art or competitive zero-shot forecasting performance, underscores its ability to perform effectively on entirely unseen tasks.

## 4.2 Forecasting under Anomalous Segments

Handling anomalous segments in time series forecasting remains a major challenge, as many SOTA models are easily misled by such irregularities, resulting in degraded performance. Our proposed TimeExpert, powered by the TMOE mechanism, can selectively exploit informative temporal segments while filtering out noisy or redundant ones, thereby improving robustness under anomalous conditions. In the following, we provide a detailed analysis of its performance across three representative anomaly cases with reference to Figure 3.

**On Abrupt Outliers** Abrupt outliers, often introduced by factors such as equipment errors, are easily misinterpreted by models as inherent trends. In (a), a sharp valley within the shaded area severely disrupts the baseline models. iTransformer incorrectly learns this drastic fluctuation, resulting in phase shifts in its predictions; for instance, at time steps 640–645 and 680–685, its forecast even moves in the opposite direction of the ground truth. PatchTST is similarly affected, with its prediction dragged downward and the overall amplitude suppressed across the forecast horizon. By comparison, TimeExpert remains largely unaffected, filtering out this transient and non-structural noise while maintaining focus on the more stable long-term dynamics. This demonstrates its robustness to abrupt but non-representative anomalies.

**On Deviation from Periodicity** When the input deviates from its inherent periodicity, models may misinterpret these fluctuations as other hidden meaningful patterns. This is evident in (c), where in the valley region between time steps 1380–1400, PatchTST replicates the anomalous signal with high-frequency spikes, while iTransformer produces overly conservative forecasts with insufficient amplitudes for both peaks and valleys. In contrast, TimeExpert emphasizes the dominant, historically stable periodic patterns, thereby preserving the overall periodic structure and producing a closer fit to the ground truth. This indicates that TimeExpert is better able to generalize when temporary deviations disrupt the underlying periodic rhythm.

**On Zero Imputation** Zero imputation, a common data artifact, introduces flat intervals that can distort model predictions. In (e), the prolonged zero-filled segment causes both PatchTST and iTransformer to underestimate the dynamic range of the future series, leading to muted forecasts for upcoming peaks. TimeExpert, however, treats the zero-filled region as uninformative and down-weights its influence, relying instead on valid patterns preceding the anomaly. This enables it to better anticipate the forthcoming high-amplitude peaks and maintain predictive accuracy. Such behavior reflects its ability to mitigate the impact of missing-data artifacts on future forecasting.

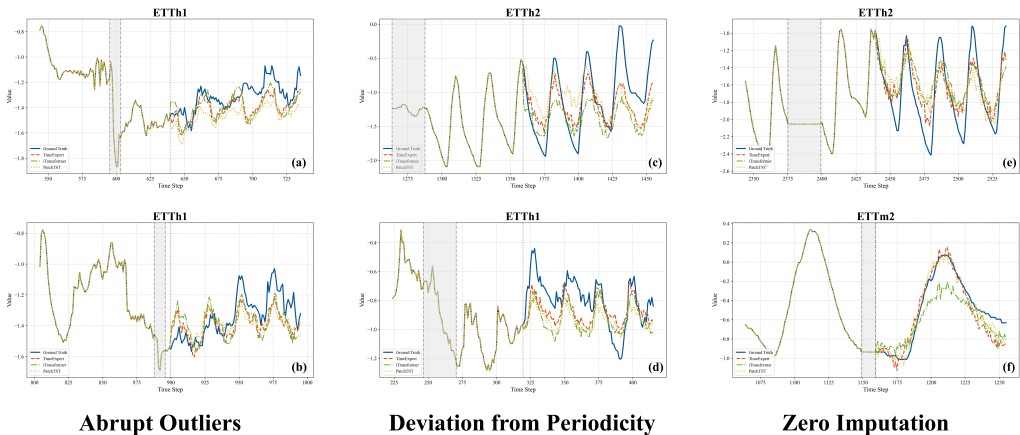

**Abrupt Outliers**  **Deviation from Periodicity**  **Zero Imputation**

Figure 3: Comparison of forecasting performance between TimeExpert, PatchTST and iTransformer on test sets with anomalous segments. The gray shaded area highlights the anomalous segment within the input sequence. The leftmost column (a, b) shows the prediction performance on sequences with "Abrupt Outliers". The middle column (c, d) demonstrates the results for sequences exhibiting "Deviation from Periodicity". The rightmost column (e, f) corresponds to cases where the input sequence contains "Zero Imputation" artifacts. In each graph, the first 96 steps represent the input sequence, and the last 96 steps show the prediction results of each model and the corresponding Ground Truth.

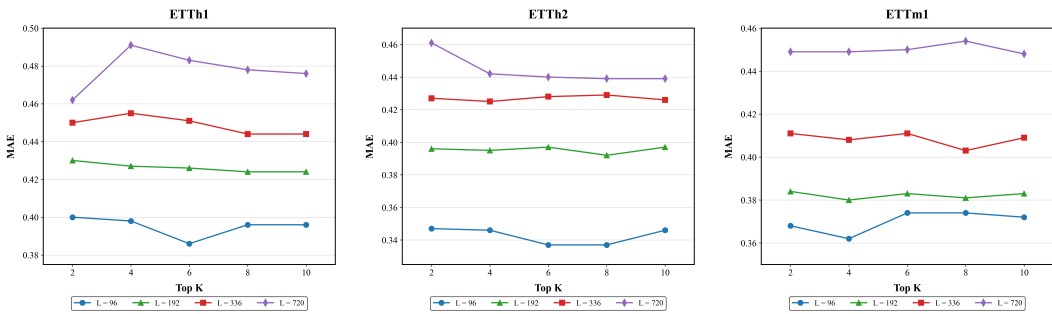

Figure 4: Influence of the selection of Top-K in different prediction length (L). We choose ETTh1, ETTh2 and ETTm1 for visualization.

Overall, these results confirm that TimeExpert consistently shows stronger resilience than baseline models across different anomaly types. By filtering out transient disturbances and redundant fluctuations through its TMOE mechanism, it preserves stable temporal dependencies and achieves more faithful forecasts under challenging anomalous conditions.

## 4.3 ABLATION STUDY

**Top-K Adaptive Local Expert Selection**  As shown in Figure 4, most datasets exhibit a distinct "U-shaped" curve with respect to K, revealing a critical trade-off. When K is too small, the model captures only a narrow temporal context, leading to underutilization of relevant information. A small K provides insufficient temporal context, while a large K introduces noise from irrelevant signals and diminishes sparsity benefits. The optimal performance at an intermediate K demonstrates that our Adaptive Local Expert Selection effectively balances context coverage against noise suppression by adaptively selecting the most informative timestamps.

**Attention Mechanisms**  To isolate the contribution of our proposed attention mechanism, we conduct an ablation study comparing three variants: Full Attention, Random Attention, and our TMOE.

Table 3: Ablation study of different attention mechanisms. The best results are in **bold**.

| Datasets | ETTh1 | | ETTh2 | | ETTm1 | | ETTm2 | | Exchange | |
|---|---|---|---|---|---|---|---|---|---|---|
| Metric | MSE | MAE | MSE | MAE | MSE | MAE | MSE | MAE | MSE | MAE |
| Full Attention | 0.438 | 0.435 | 0.374 | 0.401 | 0.390 | 0.403 | 0.284 | 0.329 | 0.364 | 0.404 |
| Random Attention | 0.433 | 0.433 | 0.379 | 0.407 | 0.392 | 0.405 | 0.285 | 0.331 | 0.369 | 0.409 |
| **TMOE** | **0.428** | **0.432** | **0.370** | **0.399** | **0.380** | **0.396** | **0.281** | **0.326** | **0.344** | **0.394** |

Table 4: Ablation study on the parameter sharing structure across ETTm2 and Solar-Energy datasets. Avg means the average results from all four prediction lengths. Best results are in **bold**.

| Pred Len | ETTm2 Dataset | | | | Solar-Energy Dataset | | | |
|---|---|---|---|---|---|---|---|---|
| | Share | | w/o Share | | Share | | w/o Share | |
| | MSE | MAE | MSE | MAE | MSE | MAE | MSE | MAE |
| 96 | **0.175** | **0.260** | 0.177 | 0.261 | **0.200** | **0.242** | 0.201 | 0.243 |
| 192 | 0.246 | 0.310 | **0.241** | **0.302** | **0.230** | **0.261** | 0.233 | 0.265 |
| 336 | **0.306** | 0.346 | **0.306** | **0.345** | **0.244** | **0.273** | 0.247 | 0.275 |
| 720 | 0.407 | 0.402 | **0.403** | **0.398** | **0.242** | **0.274** | 0.247 | 0.275 |
| Avg | 0.283 | 0.330 | **0.282** | **0.326** | **0.229** | **0.263** | 0.232 | 0.265 |

The results are presented in Table 3. The findings are conclusive: TMOE consistently and significantly outperforms both baselines across all five benchmark datasets (ETTh1, ETTh2, ETTm1, ETTm2, and Exchange) on both MSE and MAE metrics. For example, on the Exchange dataset, TMOE achieves an MSE of 0.344, which is a marked improvement over Full Attention (0.364) and Random Attention (0.369). Notably, standard Full Attention consistently outperforms Random Attention, which implies that a naive reduction in computational complexity through random sampling leads to performance degradation. In contrast, the superior results of TMOE demonstrate that its design effectively captures salient temporal patterns while mitigating the influence of spurious information, thereby yielding enhanced forecasting accuracy.

**Shared Global Expert Mechanism**   Finally, we compared configurations that include the shared global expert (Share) with those that rely solely on local experts (w/o Share). The results presented in Table 4 clearly indicate that the effectiveness of this mechanism is highly data-dependent. On datasets like ETTm2, which may contain more heterogeneous components and complex patterns, the version without shared global expert performs better, suggesting that allowing local experts to specialize in distinct patterns is more effective. On the other hand, for datasets such as Solar-Energy, which exhibit stronger homogeneity or recurring patterns, the shared global expert mechanism significantly improves performance. In this case, it enables the model to capture and utilize global, shared knowledge across sequences, thereby enhancing predictive accuracy and generalization. This observation confirms the flexibility and soundness of our model design: by integrating both shared and independent experts, TimeExpert can adaptively balance the learning of general and task-specific knowledge according to data characteristics, achieving strong performance across a wide range of forecasting tasks.

## 5   CONCLUSION

This work addresses a critical limitation of Transformer-based time series forecasting models: their rigid global context aggregation, which fails to adapt to the inherent lag effects and anomalous segments in real-world data. By reimagining the core of self-attention with the Temporal Mix of Experts (TMOE) mechanism, we propose TimeExpert and TimeExpert-G. They balance adaptive local context selection and robust global dependency capture, while retaining compatibility with standard Transformer architectures. Extensive experiments across seven diverse real-world benchmarks validate TimeExpert's effectiveness. In summary, the TMOE mechanism demonstrates a promising path to making Transformers more adaptive to real-world time series characteristics.

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

## A   APPENDIX

### A.1   USE OF LLMS

We acknowledge the use of Large Language Models (LLMs) to assist with language editing of this paper. The assistance was strictly limited to grammar checking and minor sentence-level polishing. The authors confirm that LLMs were not used for generating or altering any scientific content, interpreting data, or drawing conclusions.

### A.2   EXPERIMENTAL SETUP DETAILS

**Datasets**   We conduct extensive experiments on 7 widely used real-world datasets from diverse application domains, including ETT (ETTh1, ETTh2, ETTm1, ETTm2), Exchange, Weather, and Solar-Energy. For the experiments with TimeExpert, we follow the experimental settings in prior studies (Wu et al., 2021), where the ETT series is split into training, validation, and test sets with a 6:2:2 ratio, and a 7:1:2 split for the other datasets. For TimeExpert-G, we adopt the same dataset split settings and, in addition, follow the same pre-training settings of Timer (Liu et al., 2024b), including pre-training it on the UTSD-12G dataset with a segment length $S = 96$ and a token count $N = 15$.

**Baselines**   Our TimeExpert is fairly compared with 10 representative and state-of-the-art (SOTA) time-series forecasting methods, including (1) Mamba-based methods: TimePro (Ma et al., 2025), S-Mamba (Wang et al., 2025); (2) MLP-based methods: SOFTS (Han et al., 2024), TiDE (Das et al., 2023), DLinear (Zeng et al., 2023); (3) CNN-based methods: TimesNet (Wu et al., 2022), SCINet (Liu et al., 2022); (4) Transformer-based methods: iTransformer (Liu et al., 2024a), Crossformer (Zhang & Yan, 2023), PatchTST (Nie et al., 2023). For TimeExpert-G, we also select five foundation models for comparison in the zero-shot setting, including Timer (Liu et al., 2024b), MOIRAI (Woo et al., 2024), Chronos (Ansari et al., 2024), TimesFM (Das et al., 2024), Time-MoE (Shi et al., 2025).

**Forecasting benchmarks**   The long-term forecasting benchmarks are based on the settings of Informer and SCINet, with a lookback window length (L) of 96 and prediction horizons (H) set to [96, 192, 336, 720] for all datasets. Performance is evaluated using two primary metrics: Mean Squared Error (MSE) and Mean Absolute Error (MAE). The results are compared across different methods, with data taken from iTransformer and TimePro.

### A.3   FULL RESULTS

Table 5 and 6 show the full results of the prediction benchmarks. We conduct experiments using seven widely used real-world datasets and compare our method to other previous state-of-the-art models. The comparison models use a variety of architectures including convolution, transformer, mlp, and mamba. In these tests, our TimeExpert and TimeExpert-G achieve the best performance on most of the datasets.

We also perform more visual analyses. Our TimeExpert achieves superior results in both trend alignment and local fluctuation capture. As shown in Figure 7, TimeExpert effectively tracks peaks and troughs of the ground truth series, particularly in regions with complex volatility, where its deviations from the true values are smaller. This highlights its enhanced stability and robustness. In contrast, while PatchTST shows reasonable accuracy in capturing overall trends, it suffers from noticeable shifts at extreme points and in regions of rapid oscillation. iTransformer, on the other hand, exhibits larger deviations in both amplitude and phase alignment, leading to an overall weaker fit to the ground truth. These findings suggest that TimeExpert has stronger modeling capacity for temporal dependencies and nonlinear characteristics in time series, allowing it to better capture complex dynamic patterns. Consequently, TimeExpert demonstrates significant advantages on the ETTh2 dataset, underscoring its potential for broader practical applications.

Besides, Figure 8 illustrates the prediction comparison on the ETTm2 dataset, with a particular focus on the models' ability to capture trend changes and critical turning points. TimeExpert excels in capturing overall trends and key inflection points, as evidenced by its prediction curve closely align-

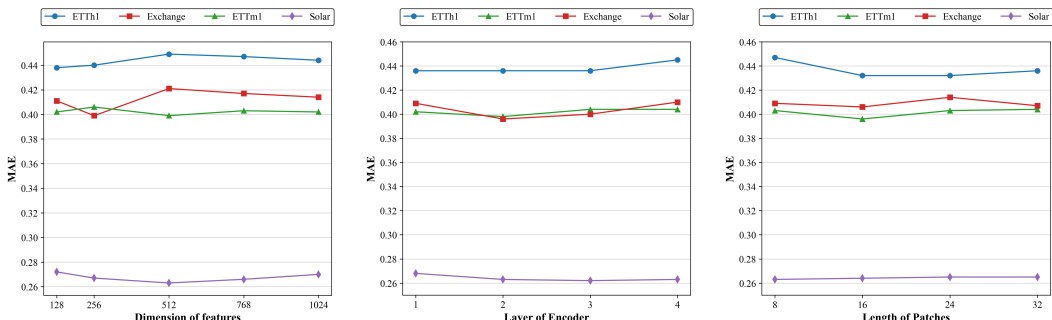

Figure 5: Influence of the hidden dimension of features, number of encoder layers and patch length. We select ETTh1, ETTm1, Exchange and Solar-Energy for visualization.

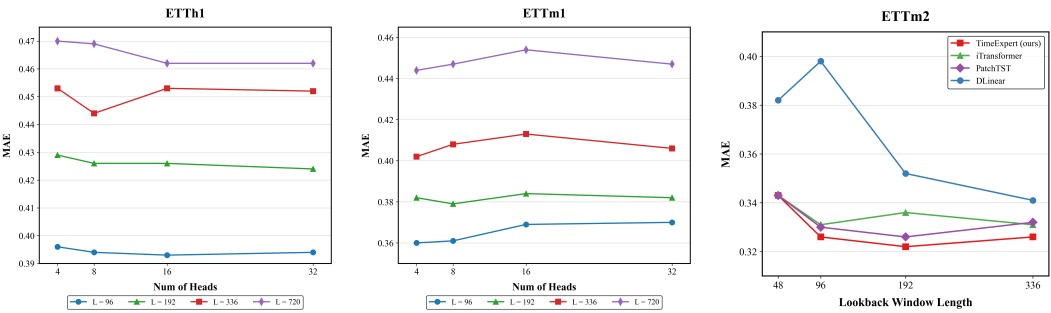

Figure 6: Ablation Studies on the Number of Heads and Lookback Window Length in Time Series Forecasting Models. The influence of number of heads is investigated using the ETTh1 and ETTm1 datasets, while the lookback window length is evaluated using the ETTm2 dataset.

ing with the ground truth during both ascending and descending phases. This model demonstrates superior accuracy in fitting peaks and troughs, reflecting its robust capability to model nonlinear dynamics within time series data. In contrast, PatchTST achieves reasonable trend accuracy but tends to lag slightly in peak regions, indicating limitations in capturing rapid local changes. iTransformer, however, shows larger deviations, particularly around turning points and segments with significant amplitude changes, which results in weaker alignment with the ground truth. Overall, TimeExpert not only offers stronger trend prediction abilities but also exhibits greater robustness in handling local fluctuations, confirming its advantages and reliability in complex time series forecasting tasks.

## A.4  MORE ABLATION RESULTS

**Feature Dimension.** Figure 5 shows that for most datasets, a feature dimension of 128 is sufficient to achieve near-optimal performance. Further increases in dimension do not result in significant improvements and may even slightly degrade performance due to parameter redundancy. This ensures TimeExpert operates efficiently with lower computational overhead.

**Number of Encoder Layers.** The model exhibits very low sensitivity to the number of encoder layers according to Figure 5. On most datasets, two encoder layers are enough to achieve optimal performance, and notably on ETTh1, even a single encoder layer achieves the best performance, underscoring the architecture's strong representational capacity to capture key features and dependencies without suffering from redundancy, overfitting, or gradient issues typical of deeper models.

**Patch Length** As presented in Figure 5, a patch size of 16 delivers the best results across most datasets. This indicates a crucial trade-off: if the patch is too short (e.g., 8), it may struggle to capture full periodic or trend patterns due to limited local context; if the patch is too long (e.g., 24 or 32), it may mix too much irrelevant information, blurring the local semantic representation. Therefore, a moderate patch size is critical for the model to effectively extract local semantics from time-series data.

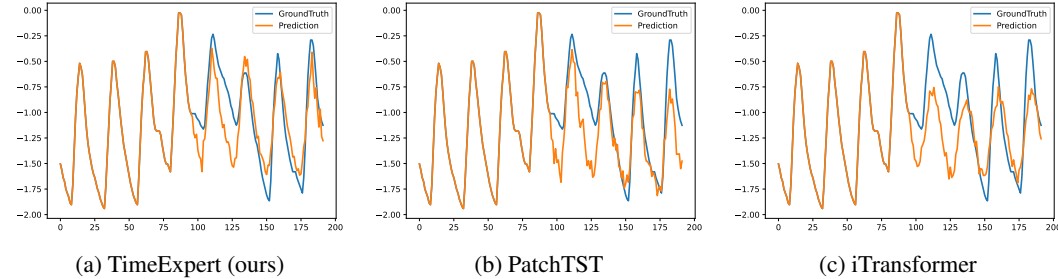

(a) TimeExpert (ours)  (b) PatchTST  (c) iTransformer

Figure 7: Comparison of forecasts between TimeExpert, PatchTST and iTransformer on ETTh2 dataset when the input length is 96 and the forecast length is 96. The blue line represents the ground truth and the orange line represents the forecast.

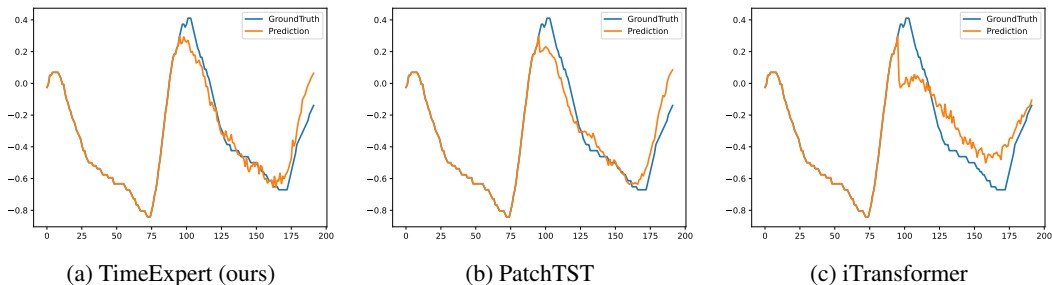

(a) TimeExpert (ours)  (b) PatchTST  (c) iTransformer

Figure 8: Comparison of forecasts between TimeExpert, PatchTST and iTransformer on ETTm2 dataset when the input length is 96 and the forecast length is 96. The blue line represents the ground truth and the orange line represents the forecast.

**Input Sequence Length** To evaluate the model's ability to handle different lengths of historical data, we conduct experiments on the ETTm2 dataset with varying lookback window lengths. Figure 6 shows that TimeExpert outperforms all baseline models consistently across all tested window lengths. Notably, with a lookback window of 96, the model's performance already reaches near-optimal levels. This is particularly significant from an engineering perspective, as it demonstrates that the model can make accurate future predictions using only a relatively short history of data, thus significantly reducing storage and computational requirements in practical applications.

**Number of Attention Heads** Since the attention module is designed in a head-wise manner, we perform an ablation study on the number of attention heads. Figure 6 show that the model is largely insensitive to this parameter, with minimal performance fluctuation across datasets and input lengths. In most cases, the optimal performance occurs with 8 or 16 heads, with negligible differences compared to other settings. This suggests that a relatively small number of heads is sufficient for Time-Expert to capture the dominant predictive patterns, reaffirming the architectural efficiency of our model.

Table 5: Full results of the multivariate long-term forecasting result comparison. The input sequence length L is set to 96 for all baselines and the prediction lengths $H \in \{96, 192, 336, 720\}$. Avg means the average results from all four prediction lengths. **Red**: the best, **Blue**: the 2nd best.

| Models | | TimeExpert (ours) | | TimePro (2025) | | S-Mamba (2025) | | SOFTS (2024) | | iTransformer (2024b) | | PatchTST (2023) | | Crossformer (2023) | | TiDE (2023) | | TimesNet (2023) | | DLinear (2023) | | SCINet (2022) | |
|---|---|---|---|---|---|---|---|---|---|---|---|---|---|---|---|---|---|---|---|---|---|---|---|
| Metric | | MSE | MAE | MSE | MAE | MSE | MAE | MSE | MAE | MSE | MAE | MSE | MAE | MSE | MAE | MSE | MAE | MSE | MAE | MSE | MAE | MSE | MAE |
| ETTm1 | 96 | 0.316 | 0.359 | 0.326 | 0.364 | 0.333 | 0.368 | 0.325 | 0.361 | 0.334 | 0.368 | 0.329 | 0.365 | 0.404 | 0.426 | 0.364 | 0.387 | 0.338 | 0.375 | 0.345 | 0.372 | 0.418 | 0.438 |
| | 192 | 0.359 | 0.379 | 0.367 | 0.383 | 0.376 | 0.390 | 0.375 | 0.389 | 0.377 | 0.391 | 0.380 | 0.394 | 0.450 | 0.451 | 0.398 | 0.404 | 0.374 | 0.387 | 0.380 | 0.389 | 0.439 | 0.450 |
| | 336 | 0.390 | 0.402 | 0.402 | 0.409 | 0.408 | 0.413 | 0.405 | 0.412 | 0.426 | 0.420 | 0.400 | 0.410 | 0.532 | 0.515 | 0.428 | 0.425 | 0.410 | 0.411 | 0.413 | 0.413 | 0.490 | 0.485 |
| | 720 | 0.454 | 0.444 | 0.469 | 0.446 | 0.475 | 0.448 | 0.466 | 0.447 | 0.491 | 0.459 | 0.475 | 0.453 | 0.666 | 0.589 | 0.487 | 0.461 | 0.478 | 0.450 | 0.474 | 0.453 | 0.595 | 0.550 |
| | Avg | 0.380 | 0.396 | 0.391 | 0.400 | 0.398 | 0.405 | 0.393 | 0.403 | 0.407 | 0.410 | 0.396 | 0.406 | 0.513 | 0.496 | 0.419 | 0.419 | 0.400 | 0.406 | 0.403 | 0.407 | 0.485 | 0.481 |
| ETTm2 | 96 | 0.175 | 0.260 | 0.178 | 0.260 | 0.179 | 0.263 | 0.180 | 0.261 | 0.180 | 0.264 | 0.184 | 0.264 | 0.287 | 0.366 | 0.207 | 0.305 | 0.187 | 0.267 | 0.193 | 0.292 | 0.286 | 0.377 |
| | 192 | 0.241 | 0.302 | 0.242 | 0.303 | 0.250 | 0.309 | 0.246 | 0.306 | 0.250 | 0.309 | 0.246 | 0.306 | 0.414 | 0.492 | 0.290 | 0.364 | 0.249 | 0.309 | 0.284 | 0.362 | 0.399 | 0.445 |
| | 336 | 0.306 | 0.345 | 0.303 | 0.342 | 0.312 | 0.349 | 0.319 | 0.352 | 0.311 | 0.348 | 0.308 | 0.346 | 0.597 | 0.542 | 0.377 | 0.422 | 0.321 | 0.351 | 0.369 | 0.427 | 0.637 | 0.591 |
| | 720 | 0.403 | 0.398 | 0.400 | 0.399 | 0.411 | 0.406 | 0.405 | 0.401 | 0.412 | 0.407 | 0.409 | 0.402 | 1.730 | 1.042 | 0.558 | 0.524 | 0.408 | 0.403 | 0.554 | 0.522 | 0.960 | 0.735 |
| | Avg | 0.281 | 0.326 | 0.281 | 0.326 | 0.288 | 0.332 | 0.287 | 0.330 | 0.288 | 0.332 | 0.287 | 0.330 | 0.757 | 0.610 | 0.358 | 0.404 | 0.291 | 0.333 | 0.350 | 0.401 | 0.571 | 0.537 |
| ETTh1 | 96 | 0.375 | 0.396 | 0.375 | 0.398 | 0.386 | 0.405 | 0.381 | 0.399 | 0.386 | 0.405 | 0.394 | 0.406 | 0.423 | 0.448 | 0.479 | 0.464 | 0.384 | 0.402 | 0.386 | 0.400 | 0.654 | 0.599 |
| | 192 | 0.420 | 0.426 | 0.427 | 0.429 | 0.443 | 0.437 | 0.435 | 0.431 | 0.441 | 0.436 | 0.440 | 0.435 | 0.471 | 0.474 | 0.525 | 0.492 | 0.436 | 0.429 | 0.437 | 0.432 | 0.719 | 0.631 |
| | 336 | 0.457 | 0.444 | 0.472 | 0.450 | 0.489 | 0.468 | 0.480 | 0.452 | 0.487 | 0.458 | 0.491 | 0.462 | 0.570 | 0.546 | 0.565 | 0.515 | 0.491 | 0.469 | 0.481 | 0.459 | 0.778 | 0.659 |
| | 720 | 0.458 | 0.462 | 0.476 | 0.474 | 0.502 | 0.489 | 0.499 | 0.488 | 0.503 | 0.491 | 0.487 | 0.479 | 0.653 | 0.621 | 0.594 | 0.558 | 0.521 | 0.500 | 0.519 | 0.516 | 0.836 | 0.699 |
| | Avg | 0.428 | 0.432 | 0.438 | 0.438 | 0.455 | 0.450 | 0.449 | 0.442 | 0.454 | 0.447 | 0.453 | 0.446 | 0.529 | 0.522 | 0.541 | 0.507 | 0.458 | 0.450 | 0.456 | 0.452 | 0.747 | 0.647 |
| ETTh2 | 96 | 0.288 | 0.337 | 0.293 | 0.345 | 0.296 | 0.348 | 0.297 | 0.347 | 0.297 | 0.349 | 0.288 | 0.340 | 0.745 | 0.584 | 0.400 | 0.440 | 0.340 | 0.374 | 0.333 | 0.387 | 0.707 | 0.621 |
| | 192 | 0.365 | 0.392 | 0.367 | 0.394 | 0.376 | 0.396 | 0.373 | 0.394 | 0.380 | 0.400 | 0.376 | 0.395 | 0.877 | 0.656 | 0.528 | 0.509 | 0.402 | 0.414 | 0.477 | 0.476 | 0.860 | 0.689 |
| | 336 | 0.408 | 0.425 | 0.419 | 0.431 | 0.424 | 0.431 | 0.410 | 0.426 | 0.428 | 0.432 | 0.440 | 0.451 | 1.043 | 0.731 | 0.643 | 0.571 | 0.452 | 0.452 | 0.594 | 0.541 | 1.000 | 0.744 |
| | 720 | 0.420 | 0.439 | 0.427 | 0.445 | 0.426 | 0.444 | 0.411 | 0.433 | 0.427 | 0.445 | 0.436 | 0.453 | 1.104 | 0.763 | 0.874 | 0.679 | 0.462 | 0.468 | 0.831 | 0.657 | 1.249 | 0.838 |
| | Avg | 0.370 | 0.399 | 0.377 | 0.403 | 0.381 | 0.405 | 0.373 | 0.400 | 0.383 | 0.407 | 0.385 | 0.410 | 0.942 | 0.684 | 0.611 | 0.550 | 0.414 | 0.427 | 0.559 | 0.515 | 0.954 | 0.723 |
| Exchange | 96 | 0.084 | 0.201 | 0.085 | 0.204 | 0.086 | 0.207 | 0.091 | 0.209 | 0.086 | 0.206 | 0.088 | 0.205 | 0.256 | 0.367 | 0.094 | 0.218 | 0.107 | 0.234 | 0.088 | 0.218 | 0.267 | 0.396 |
| | 192 | 0.171 | 0.294 | 0.178 | 0.299 | 0.182 | 0.304 | 0.176 | 0.303 | 0.177 | 0.299 | 0.176 | 0.299 | 0.470 | 0.509 | 0.184 | 0.307 | 0.226 | 0.344 | 0.176 | 0.315 | 0.351 | 0.459 |
| | 336 | 0.316 | 0.406 | 0.328 | 0.414 | 0.332 | 0.418 | 0.329 | 0.416 | 0.331 | 0.417 | 0.301 | 0.397 | 1.268 | 0.883 | 0.349 | 0.431 | 0.367 | 0.448 | 0.313 | 0.427 | 1.324 | 0.853 |
| | 720 | 0.808 | 0.673 | 0.817 | 0.679 | 0.867 | 0.703 | 0.848 | 0.680 | 0.847 | 0.691 | 0.901 | 0.714 | 1.767 | 1.068 | 0.852 | 0.698 | 0.964 | 0.746 | 0.839 | 0.695 | 1.058 | 0.797 |
| | Avg | 0.344 | 0.394 | 0.352 | 0.399 | 0.367 | 0.408 | 0.361 | 0.402 | 0.360 | 0.403 | 0.367 | 0.404 | 0.940 | 0.707 | 0.370 | 0.413 | 0.416 | 0.443 | 0.354 | 0.414 | 0.750 | 0.626 |
| Weather | 96 | 0.168 | 0.213 | 0.166 | 0.207 | 0.165 | 0.210 | 0.166 | 0.208 | 0.174 | 0.214 | 0.176 | 0.217 | 0.158 | 0.230 | 0.202 | 0.261 | 0.172 | 0.220 | 0.196 | 0.255 | 0.221 | 0.306 |
| | 192 | 0.216 | 0.253 | 0.216 | 0.254 | 0.214 | 0.252 | 0.217 | 0.253 | 0.221 | 0.254 | 0.221 | 0.256 | 0.206 | 0.277 | 0.242 | 0.298 | 0.219 | 0.261 | 0.237 | 0.296 | 0.261 | 0.340 |
| | 336 | 0.271 | 0.295 | 0.273 | 0.296 | 0.274 | 0.297 | 0.282 | 0.300 | 0.278 | 0.296 | 0.275 | 0.296 | 0.272 | 0.335 | 0.287 | 0.335 | 0.280 | 0.306 | 0.283 | 0.335 | 0.309 | 0.378 |
| | 720 | 0.348 | 0.344 | 0.351 | 0.346 | 0.350 | 0.345 | 0.356 | 0.351 | 0.358 | 0.347 | 0.352 | 0.346 | 0.398 | 0.418 | 0.351 | 0.386 | 0.365 | 0.359 | 0.345 | 0.381 | 0.377 | 0.427 |
| | Avg | 0.251 | 0.276 | 0.251 | 0.276 | 0.251 | 0.276 | 0.255 | 0.278 | 0.258 | 0.278 | 0.256 | 0.279 | 0.259 | 0.315 | 0.271 | 0.320 | 0.259 | 0.287 | 0.265 | 0.317 | 0.292 | 0.363 |
| Solar-Energy | 96 | 0.200 | 0.242 | 0.196 | 0.237 | 0.205 | 0.244 | 0.200 | 0.230 | 0.203 | 0.237 | 0.205 | 0.246 | 0.310 | 0.331 | 0.312 | 0.399 | 0.250 | 0.292 | 0.290 | 0.378 | 0.237 | 0.344 |
| | 192 | 0.229 | 0.261 | 0.231 | 0.263 | 0.237 | 0.270 | 0.229 | 0.253 | 0.233 | 0.261 | 0.237 | 0.267 | 0.734 | 0.725 | 0.339 | 0.416 | 0.296 | 0.318 | 0.320 | 0.398 | 0.280 | 0.380 |
| | 336 | 0.243 | 0.273 | 0.250 | 0.281 | 0.258 | 0.288 | 0.243 | 0.269 | 0.248 | 0.273 | 0.250 | 0.276 | 0.750 | 0.735 | 0.368 | 0.430 | 0.319 | 0.330 | 0.353 | 0.415 | 0.304 | 0.389 |
| | 720 | 0.242 | 0.274 | 0.253 | 0.285 | 0.260 | 0.288 | 0.245 | 0.272 | 0.249 | 0.275 | 0.252 | 0.275 | 0.769 | 0.765 | 0.370 | 0.425 | 0.338 | 0.337 | 0.356 | 0.413 | 0.308 | 0.388 |
| | Avg | 0.229 | 0.263 | 0.232 | 0.266 | 0.240 | 0.273 | 0.229 | 0.256 | 0.233 | 0.262 | 0.236 | 0.266 | 0.641 | 0.639 | 0.347 | 0.417 | 0.301 | 0.319 | 0.330 | 0.401 | 0.282 | 0.375 |

Table 6: Full results of zero-shot forecasting experiments. Lower MSE or MAE values indicate better predictions. A dash ('-') denotes datasets included in the model's pretraining and therefore excluded from testing. **Red**: the best, Blue: the 2nd best.

| Models | TimeExpert-G | | Timer | | MOIRAI | | Chronos | | TimesFM | | Time-MoE | |
|---|---|---|---|---|---|---|---|---|---|---|---|---|
| Metric | MSE | MAE | MSE | MAE | MSE | MAE | MSE | MAE | MSE | MAE | MSE | MAE |
| ETTm1 96 | **0.301** | **0.357** | 0.698 | 0.530 | 0.353 | 0.363 | 0.402 | 0.373 | 0.363 | 0.369 | 0.309 | **0.357** |
| ETTm1 192 | **0.330** | **0.378** | 0.744 | 0.555 | 0.376 | 0.380 | 0.510 | 0.435 | 0.417 | 0.405 | 0.346 | 0.381 |
| ETTm1 336 | **0.358** | **0.392** | 0.801 | 0.582 | 0.399 | 0.395 | 0.590 | 0.477 | 0.447 | 0.428 | 0.373 | 0.408 |
| ETTm1 720 | **0.392** | 0.421 | 0.829 | 0.606 | 0.432 | **0.417** | 0.703 | 0.525 | 0.513 | 0.470 | 0.475 | 0.477 |
| ETTm1 avg | **0.345** | **0.387** | 0.768 | 0.568 | 0.390 | 0.389 | 0.551 | 0.453 | 0.435 | 0.418 | 0.376 | 0.406 |
| ETTm2 96 | **0.189** | 0.277 | 0.225 | 0.300 | **0.189** | **0.260** | 0.192 | 0.263 | 0.206 | 0.267 | 0.197 | 0.286 |
| ETTm2 192 | **0.237** | 0.314 | 0.286 | 0.339 | 0.247 | **0.300** | 0.256 | 0.308 | 0.293 | 0.320 | 0.250 | 0.322 |
| ETTm2 336 | **0.286** | 0.354 | 0.335 | 0.369 | 0.295 | **0.334** | 0.315 | 0.346 | 0.411 | 0.414 | 0.337 | 0.375 |
| ETTm2 720 | 0.346 | 0.398 | 0.414 | 0.416 | 0.372 | **0.386** | 0.409 | 0.405 | 0.478 | 0.437 | 0.475 | 0.477 |
| ETTm2 avg | **0.265** | 0.336 | 0.315 | 0.356 | 0.276 | 0.320 | 0.293 | **0.331** | 0.347 | 0.360 | 0.315 | 0.365 |
| ETTh1 96 | **0.338** | **0.378** | 0.454 | 0.434 | 0.380 | 0.398 | 0.389 | 0.409 | 0.421 | 0.401 | 0.350 | 0.382 |
| ETTh1 192 | **0.381** | **0.406** | 0.522 | 0.465 | 0.440 | 0.434 | 0.502 | 0.443 | 0.472 | 0.432 | 0.388 | 0.412 |
| ETTh1 336 | **0.402** | **0.428** | 0.559 | 0.484 | 0.514 | 0.474 | 0.580 | 0.460 | 0.510 | 0.455 | 0.411 | 0.430 |
| ETTh1 720 | **0.424** | 0.457 | 0.714 | 0.549 | 0.705 | 0.568 | 0.605 | 0.495 | 0.514 | 0.481 | 0.427 | **0.455** |
| ETTh1 avg | **0.386** | 0.417 | 0.562 | 0.483 | 0.510 | 0.469 | 0.519 | 0.452 | 0.479 | 0.442 | 0.394 | 0.420 |
| ETTh2 96 | **0.277** | 0.336 | 0.316 | 0.359 | 0.287 | **0.325** | 0.306 | 0.338 | 0.326 | 0.355 | 0.302 | 0.354 |
| ETTh2 192 | **0.335** | 0.381 | 0.374 | 0.398 | 0.347 | **0.367** | 0.396 | 0.394 | 0.397 | 0.400 | 0.364 | 0.385 |
| ETTh2 336 | **0.360** | 0.408 | 0.381 | 0.410 | 0.377 | **0.393** | 0.423 | 0.417 | 0.431 | 0.413 | 0.417 | 0.425 |
| ETTh2 720 | 0.411 | 0.441 | 0.408 | 0.434 | **0.404** | **0.421** | 0.442 | 0.439 | 0.446 | 0.444 | 0.527 | 0.496 |
| ETTh2 avg | **0.346** | 0.392 | 0.370 | 0.400 | 0.354 | **0.377** | 0.392 | 0.397 | 0.400 | 0.403 | 0.403 | 0.415 |
| Weather 96 | **0.140** | **0.189** | 0.190 | 0.236 | 0.177 | 0.208 | 0.186 | 0.208 | - | - | 0.159 | 0.213 |
| Weather 192 | **0.201** | **0.242** | 0.261 | 0.293 | 0.219 | 0.249 | 0.238 | 0.258 | - | - | 0.215 | 0.266 |
| Weather 336 | **0.229** | **0.278** | 0.332 | 0.340 | 0.277 | 0.292 | 0.313 | 0.353 | - | - | 0.291 | 0.322 |
| Weather 720 | **0.279** | **0.314** | 0.385 | 0.381 | 0.365 | 0.350 | 0.416 | 0.415 | - | - | 0.415 | 0.400 |
| Weather avg | **0.212** | **0.256** | 0.292 | 0.313 | 0.260 | 0.275 | 0.288 | 0.309 | - | - | 0.270 | 0.300 |
| Exchange 96 | **0.086** | **0.206** | 0.095 | 0.219 | 0.096 | 0.213 | 0.099 | 0.219 | 0.096 | 0.215 | 0.145 | 0.267 |
| Exchange 192 | **0.168** | **0.293** | 0.198 | 0.322 | 0.197 | 0.312 | 0.194 | 0.314 | 0.195 | 0.313 | 0.333 | 0.397 |
| Exchange 336 | **0.302** | **0.398** | 0.349 | 0.431 | 0.349 | 0.425 | 0.341 | 0.423 | 0.332 | 0.416 | 0.367 | 0.439 |
| Exchange 720 | **0.813** | **0.678** | 0.927 | 0.729 | 0.903 | 0.717 | 0.846 | 0.690 | 0.935 | 0.723 | 0.882 | 0.712 |
| Exchange avg | **0.342** | **0.394** | 0.392 | 0.425 | 0.386 | 0.417 | 0.370 | 0.412 | 0.390 | 0.417 | 0.432 | 0.454 |
| Solar-Energy 96 | **0.280** | 0.359 | 0.591 | 0.504 | 0.682 | 0.688 | 0.373 | **0.304** | 0.408 | 0.345 | 0.304 | 0.345 |
| Solar-Energy 192 | 0.366 | 0.380 | 0.689 | 0.567 | 0.694 | 0.695 | 0.363 | **0.303** | 0.466 | 0.373 | 0.309 | 0.342 |
| Solar-Energy 336 | 0.393 | 0.378 | 0.831 | 0.636 | 0.719 | 0.706 | **0.391** | 0.319 | 0.526 | 0.407 | 0.433 | 0.450 |
| Solar-Energy 720 | **0.394** | 0.374 | 0.972 | 0.710 | 0.759 | 0.725 | 0.444 | **0.349** | 0.601 | 0.461 | 0.599 | 0.576 |
| Solar-Energy avg | **0.358** | 0.372 | 0.771 | 0.604 | 0.714 | 0.704 | 0.393 | **0.319** | 0.500 | 0.397 | 0.411 | 0.428 |

