# OpenReview forum: "TimeExpert: Boosting Long Time Series Forecasting with Temporal Mix of Experts"
_ICLR.cc/2026/Conference — ICLR 2026 Conference Withdrawn Submission_

### Official Review · Reviewer_wKoc · 2025-10-25

**Soundness:** 2
**Presentation:** 3
**Contribution:** 2
**Rating:** 2
**Confidence:** 4

**Summary:**

This paper proposes a novel attention-level mechanism (TMoE) to replace the original attention mechanism in Transformer-based forecasting models. TMoE consists of several local experts and a shared global expert, to address the limitations of rigid global attention in handling lag effects and anomalies.

**Strengths:**

1. The manuscript is well-written, and the proposed methodology is presented with overall clarity.

2. The experiments show that TimeExpert achieves a competitive edge, outperforming several recent state-of-the-art baselines.

**Weaknesses:**

1. The primary mechanism of TMoE involves using the top-k indices from each row of the attention map to select the top-k experts. How does this fundamentally differ from the previously common approach of sparsifying attention via top-k selection per row (as in Informer [1])? It appears to be a similar concept but repackaged with MoE.

2. The experimental analysis lacks depth. The authors claim that TMoE can selectively exploit informative temporal segments while filtering out noisy or redundant ones. However, they do not provide visual experiments to support this argument. For instance, in the prediction visualization of Figure 3, which specific local experts are selected for each case? What patterns do they correspond to? Is there actual evidence that noisy time steps are filtered out? Figure 3 alone is insufficient to substantiate these points.

3. The evaluation omits some commonly used benchmarks, such as the Traffic and Electricity datasets. The rationale for excluding these particular datasets should be clarified, especially considering their prevalence in the literature.

[1] Zhou H, Zhang S, Peng J, et al. Informer: Beyond efficient transformer for long sequence time-series forecasting[C]//Proceedings of the AAAI conference on artificial intelligence. 2021, 35(12): 11106-11115.

**Questions:**

Please see weakness.

---

### Official Review · Reviewer_GDHp · 2025-10-31

**Soundness:** 2
**Presentation:** 2
**Contribution:** 2
**Rating:** 2
**Confidence:** 4

**Summary:**

The manuscript proposes a sparse attention mechanism, HEA, built on temporal relevance between time steps, and claims that it can suppress outliers. A forecasting framework, TimeExpert, is then constructed around HEA and is reported to outperform baseline methods.

**Strengths:**

The proposed TimeExpert model is evaluated on benchmark datasets and reports promising accuracy compared with recent state-of-the-art baselines. Its HEA module reduces attention complexity to $O(kn)$ by adaptively focusing on temporally related time steps, thereby avoiding computations on less relevant positions and improving efficiency.

**Weaknesses:**

The novelty of the proposed HEA is not clearly established. It appears to follow prior sparse-attention designs (e.g., Informer) by attending each query to a selected subset of keys, but the key-selection procedure itself is insufficiently specified. In particular, the adaptive rule in Eq. (2) does not capture locality and seasonality simultaneously.

In addition, the embedding pipeline is not described, so the exact input representation fed into HEA is unclear, making it difficult to verify what temporal or cross-variable information is captured.

The core contribution of this manuscript is to propose a sparse attention mechanism, but it does not provide any efficiency analysis to demonstrate the computational benefits over full attention or other sparse variants.

**Questions:**

I have a few suggestions which may improve the quality of the paper:

(a) The “local expert” is defined as the key–value pair at time step s, but the selection and subsequent computation do not clearly highlight why this pair is an “expert.” The procedure essentially follows standard attention (query–key productions followed by a weighted sum over values). Please clarify what is unique about the expert selection and how it changes the computation.

(b) The outlier-suppression argument is not fully convincing. If $x_s$ is an outlier, its key $k_s$ will be dissimilar to $q_t$, so a standard full-attention mechanism would also assign a low weight to $v_s$. Please justify why the additional feature-similarity term is necessary and quantify its benefit.

(c) The temporal relevance term $|t - s|$ is defined via absolute time-step distance. For local patterns (e.g., Exchange, Figure 1(a)) $|t - s|$is small; for seasonal patterns (e.g., ETTm1, Weather, Solar-Energy) time steps that are far apart in index can still be strongly related. Explain how $|t - s|$ captures both locality and seasonal recurrence, or extend the formulation to handle seasonal lags explicitly.

(d) Please explain $x_{t,d}$ in Eq. (3). What is the role of the feature index $d$?

(e) In the ablation study, add comparisons with several recent sparse-attention mechanisms for time-series forecasting to better demonstrate the effectiveness of HEA.

(f) The computation for the embedding stage and the output layer of TimeExpert is omitted from Figure 2. Please describe the processing steps before the encoder and after the encoder to make the pipeline complete.

(g) Include a quantitative comparison to demonstrate the computational advantages, reporting parameters, FLOPs, and peak memory usage.

---

### Official Review · Reviewer_EoMT · 2025-10-31

**Soundness:** 3
**Presentation:** 2
**Contribution:** 2
**Rating:** 4
**Confidence:** 4

**Summary:**

This paper introduces the Temporal Mix of Experts (TMOE), a attention-level mechanism designed to address the limitations of standard Transformers in time series forecasting. TMOE reimagines KV pairs as "local experts" and employs an adaptive, top-k selection process to filter out irrelevant or anomalous timestamps, thus handling dynamic lag effects and noise.

By integrating TMOE into existing frameworks (TimeExpert and TimeExpert-G), the authors demonstrate SOTA performance on seven real-world long-term forecasting benchmarks.

**Strengths:**

1. While there have been various attempts to capture temporal dependencies in time-series attention, the proposed Local Expert system based on a MOE framework is a novel and original approach to the problem.

2. This mechanism leads to performance improvements. The paper demonstrates that the proposed TimeExpert models achieve SOTA results across seven real-world forecasting benchmarks, validating the effectiveness of the TMOE design.

**Weaknesses:**

1. The justification for framing the mechanism as "MOE" is somewhat unclear. The core mechanism could be interpreted as a standard attention model modified with top-k filtering and an additional learned factor for temporal proximity, rather than a true Mixture-of-Experts architecture. The paper could benefit from a clearer distinction.

2. The paper claims that TimeExpert excels at filtering noise and "*PatchTST is similarly affected, with its prediction dragged downward and the overall amplitude suppressed across the forecast horizon. By comparison, TimeExpert remains largely unaffected, filtering out this transient and non-structural noise while maintaining focus on the more stable long-term dynamics.*" (Section 4.2). However, this claim is substantiated only with experiments using a short lookback window of $L=96$. This experimental setup is insufficient to validate claims about "long-term" dynamics. To properly support its central hypothesis, the paper must include experiments with much longer input sequences (e.g., $L=336$ or $L=512$ like PatchTST) to demonstrate that the TMOE mechanism can indeed identify and utilize stable long-term patterns over extended historical data, rather than just performing well on short-term contexts.

3. The paper lacks crucial qualitative analysis or visualizations to demonstrate how the Local Experts actually operate. There are no figures or analysis (e.g., heatmaps of selected indices, histograms of temporal distances) showing which experts are being selected by the TMOE mechanism. This omission makes it difficult to verify the central claims that the model is actually filtering anomalies or adaptively selecting relevant temporal lags, as opposed to just achieving good performance through other means.

**Questions:**

Please refer to weaknesses

---

### Official Review · Reviewer_uwnJ · 2025-11-01

**Soundness:** 2
**Presentation:** 3
**Contribution:** 2
**Rating:** 2
**Confidence:** 3

**Summary:**

The authors propose a mixture of expert model for time series data and evaluate it on seven standard benchmarks, improving against various baselines.

**Strengths:**

- good motivation
- well written paper
- reasonable evaluation

**Weaknesses:**

- unclear / incomplete description of MoE mechanisms, especially how it adapts to different data characteristics and temporal patterns
- good examples to show some of the main ideas but no examples showing the limits of the proposed method
- given that the results are data-dependent, how to adapt to this dependence is left unclear

**Questions:**

Treating key-value pairs as experts is quite unconventional and seem too fine granular. How does your MoE architecture compare to conventional LLM MoE architectures, and how do you select the number of KV pairs to catch different temporal patterns and lags? How do you adapt to different dataset characteristics?

---

### Note · Authors · 2025-11-19

I have read and agree with the venue's withdrawal policy on behalf of myself and my co-authors.